# An Engineering-Problem-Based Short Experiment Project on Finite Element Method for Undergraduate Students

**Yanjie Guo [1], Lijuan Yang [1], Xuefeng Chen [1] and Lei Yang [2],***

[1] School of Mechanical Engineering, Xi'an Jiaotong University, Xi'an 710049, China

[2] Key Laboratory of Education Ministry for Modern Design and Rotor-Bearing System, Xi'an Jiaotong University, Xi'an 710049, China

* Correspondence: yanglxjtu@mail.xjtu.edu.cn; Tel.: (+86-029-82669151 (F.L.)

**Abstract:** How to cultivate students' engineering practice ability in the digital manufacturing period is a difficult problem. This paper presents a finite element analysis experiment project for undergraduate students. The goals of implementing finite element modeling in this project are to enhance students' understanding of theoretical mechanics and to introduce computational methods to undergraduate students at an early stage during their undergraduate education. The students are required to use different methods to analyze the stress state of a typical engineering structure and seek the optimal design. Then, the experiment score is divided into two parts according to the course objectives. We analyzed the score distribution to find the limitations of teaching. The feedback from the students demonstrate that the experiment case from the engineering project can increase their interest in learning and applying the acquired knowledge. After comparisons, our conclusion is that outcome-based education (OBE) can effectively improve the quality of classroom teaching, and this experimental project can improve students' engineering practice ability and obtain satisfactory teaching results.

**Keywords:** outcome-based education; engineering case; result analysis

## 1. Introduction

The finite element (FE) analysis is a part of an overall design cycle in the field of computer-aided engineering (CAE) including computer-aided design and testing [1–3]. Public concern is focused on how best to improve 3E (emerging engineering education) education in China in order to increase the nation's competitiveness [4,5]. The finite-element method is developed due to the need for solving complex and large structural problems encountered in real life. This determines that the finite element is a bridge between applied mathematics and engineering [6–8].

With the rapid development of numerical manufacturing, cultivating students' ability to solve complex engineering problems using the FE method and other numerical calculations becomes more challenging [9–11]. Throughout their undergraduate studies, finite element calculations are widely used in various courses, including structural mechanics, theoretical mechanics, mechanical design basis, etc., but most of the courses focus on the teaching of theoretical knowledge. For instance, the University of Pittsburgh combined finite element analysis with physics, and let students analyze mechanical behavior during skydiving through computer programming [12]. The University of Virginia developed a set of mechanics teaching test benches to simulate the stress distribution of solid materials subject to external load [13]. However, most of the present teaching methods lack an engineering background [13–15]. Students are not aware of the application process of finite element analysis in engineering [16–18]. How to introduce engineering reality into the teaching process of finite element analysis is still a problem [19–24].

There are two main issues for the application of finite element analysis in engineering: (1) the accuracy of the analysis and (2) how to use the calculation results to optimize the design. Many of the existing finite elements courses remain at the analytical stage. This paper designed an experiment process based on the aforementioned two main problems. The process is divided into three steps—measurement, analysis, and optimization—to grasp the application of a finite element in engineering. Then, in order to accurately evaluate the learning effect for students and to improve the teaching methods, the course outcome achievement scale is evaluated.

This remainder of this paper is organized as follows: Section 2 presents the design of the teaching system and experiment plan, and the evaluation method of the course outcome is designed according to the outcome-based education (OBE) system. Section 3 the results of the students' evaluation and satisfaction. Finally, we summarize the accomplishments achieved at this stage of the experiment and offer suggestions to the next step to further improve the proposed teaching method in Section 4.

## 2. Overview of the Finite Element Experiment

### 2.1. Teaching Objectives of this Course

OBE is a structural model for organizing, implementing, and evaluating education based on an expected learning output. The OBE education model is centered on "defining expected learning output - realizing expected learning output - evaluating learning output" [25]. OBE is adopted in this course since the approach enables the articulation between education and training, recognition of prior learning, and thus the increased mobility of the students [26–31].

The schematic of the OBE-based experimental teaching process is shown in Figure 1. We can see that the course objectives are designed on the basis of Bloom's taxonomy and then supported by the experimental content and teaching methods. We use two methods to evaluate the course objective achievement: the direct evaluation method based on student score, and the indirect evaluation method based on student questionnaires. Finally, the evaluation results are analyzed, and the feedback is used for the continuous improvement of the course.

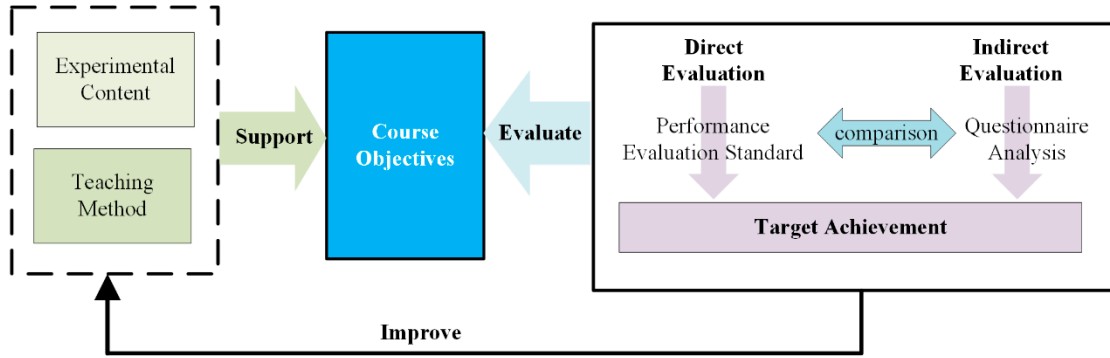

**Figure 1.** The OBE-based experiment course teaching, evaluation, and improvement process.

The main objective of this course is to train the students to solve complex engineering problems that involve digital thinking and numerical analysis. The main teaching process of the course includes classroom teaching, computer practice, extracurricular discussion, and real-engineering-problem-driven experimental teaching. The objectives and requirements of the course are as follows:

1.  Students are required to have a good understanding of the basic principles of finite element, the construction of shape functions, the selection of element types, and various numerical methods and to be able to use computer programs and finite element software to realize large-scale finite element calculation.
2.  Students should carry out effective experimental research and have the basic ability to solve complex problems in practical engineering.

## 2.2. The Process of Experiment

This experiment is meant to familiarize students with computational methods and measuring tools of stress and hence help students to apply finite element analysis (FEA) theoretical knowledge into practice. Before introducing the experiment, the basic principle of finite element method and the operation of finite element method (FEM) software (ANSYS Workbench 15.0) was taught to the students. The students mastered the calculation method, finite element node division, and total rigid matrix composition. This course requires knowledge of linear algebra, higher mathematics, numerical analysis, and theoretical mechanics. This course is meant to be taught in the first semester of the junior year for students majoring in Mechanical Engineering. The object of the finite element analysis of this experiment originates from the case of engineering practice. Its original model is derived from the autoclave opening structure. The test model built in the experiment is a structure scaled down by 1:2.5. The structure of the experiment containing a sleeve, shaft, rocker arm, and foundation is shown in Figure 2. The process of the experiment is shown in Figure 2.

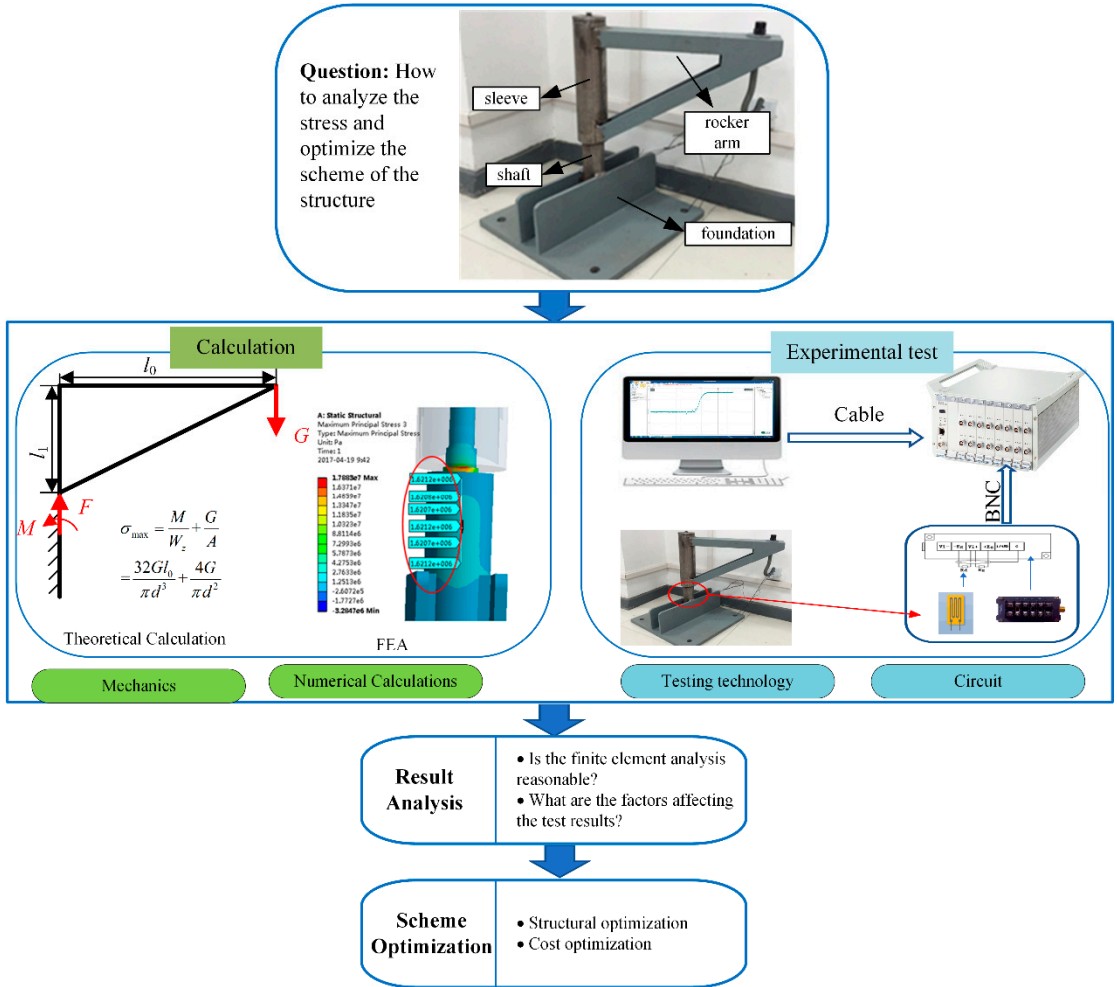

**Figure 2.** The process of the experiment.

For the typical structure of the autoclave, there is a stress concentration region during the actual working condition, where a bending fracture can easily occur. Our new experiment project is developed based on this engineering problem. The students are required to measure the stress at the stress concentration point by the experimental setup. Then, the students need to build a finite element model for the shaft and calculate the stress distribution of the shaft under the experimental loading conditions. The FEA calculation results are validated by the experimental results. Then, the students are required to propose an improved design of the shaft structure to decrease the concentration stress

based on the above results. The optimized shaft structure should be modeled again, and the stress distribution is calculated to prove the effectiveness of the proposed optimization design.

The whole experiment is divided into four steps, and the specific process of each step is described as follows.

Step 1: Structural mechanics analysis

The simplified mechanics model of the autoclave opening structure is extracted based on structural mechanics and theoretical mechanics theory. At the end of the rocker arm, we hook the weight *G* and calculate the stress.

$$\begin{cases} M = Gl_0 \\ F = G \end{cases} \Rightarrow \sigma = \frac{M}{W_z} = \frac{32Gl_0}{\pi d^3} \tag{1}$$

where $W_z$ is the coefficient of bending cross-section, d is the shaft diameter, and l0 is the length of the rocker arm，$l_0$ = 0.445 m.

Step 2: Experimental test

The stress testing apparatus consists of a data acquisition system, strain adapters, and strain sensors, as shown in Figure 2. The strain is measured using the strain gauge attached on the shaft. The output signal of the strain gauge is measured by a Wheatstone bridge. We provide students with six different measurement connection circuits for them to choose from.

First, the foil gauge is attached to the shaft. Second, the test system is connected to the chosen circuit input port. Then, the measurement software is set to measure the stress where measurement settings should be consistent with the connection method. Finally, we enforce the load and measure the strain at the designated locations. During the measuring process, the students must choose two different measuring methods and apply at least three different weights.

Step 3: The structure analysis using FEM software

Then, the 3D model of the autoclave opening structure is built by 3D drawing software such as Inventor 2016. The size is measured by the students, and they need to simplify the structure to improve the accuracy and speed of the calculation. Students can analyze the stress of the test subject through ANSYS Workbench 15.0. The results obtained by the simulation should be compared with the test results obtained in Step 2.

Step 4: The results analysis and structure optimization

The calculated result is compared with the measured result shown in Table 1, and the error is 1.85%, proving that the FE result is consistent with the strain gauge measuring result. Through the comparison, the students directly observe that the FE can accurately calculate the stress and strain of the mechanical structures. Although a small difference is observed between the numerical calculation and experiment test, the numerical calculation is still useful to reduce the number of experiments and aid structure design.

**Table 1.** Result comparisons.

| The stress in the vertical direction | | Major principal stress | |
|---|---|---|---|
| Theoretical Calculation Value（MPa） | 1.621 | FEM Calculation Value（MPa） | 1.621 |
| Test Value（MPa） | 1.651 | Test Value（MPa） | 1.675 |
| Error (%) | 1.85 | Error (%) | 3.33 |

*2.3. Evaluation Approaches*

In order to quantitatively judge each specific step of the experiment, the evaluation system of experimental results is given. We have specifically divided the completion level of each step in the experiment and gave the specific score. The specific scores for each step are as listed follows:

Step 1: Structural mechanics analysis

1. Simplify the mechanics model and calculate the stress and strain using theoretical mechanics. (2.5 points)
- 2.5 points: Calculate the right answer on your own using theoretical mechanics.
- 1.25 points: Calculate the right answer under the guidance of the teacher using theoretical mechanics.

Step 2: Experimental test

1) Standardization of equipment operation (25 points)

- 25 points: Do the experiment independently.
- 22.5 points: Complete the experiment under the guidance of the teacher.
- 18.5 points: Complete the experiment with the help of the teacher.
- 12.5 points: Master the experimental method but do not get the correct testing results.

2) Selection of the experimental method (10 points)

- 10 points: Complete two basic measurement circuits.
- 5 points: Complete one basic measurement circuits.
- +1.25 points: Using strain flowmetry.

Step 3: The structure analysis using FE software

1) Capability of problem analysis (10 points)

- 10 points: Establish a finite element model of the experiment object that is accurate and effective.
- 8.5 points: Establish a finite element model of the experiment object that can be identified in the finite element software.
- 7.5 points: Establish a finite element model of the experiment object that cannot be identified in the finite element software.
- 6 points: Use the finite element software and get reasonable results with the help of the teacher.

2) Use modern information technology and tools (12.5 points)

- 12.5 points: Perform an analysis using the finite element software and obtain reasonable results.
- 9 points: Perform an analysis using the finite element software and obtain reasonable results by adjusting.
- 7.5 points: Use the finite element software and obtain reasonable results with the guidance of the teacher.
- 6 points: Use the finite element software and obtain reasonable results with the help of the teacher.

Step 4: The result analysis and structure optimization

1) Structure optimization (12.5 points)

- 12.5 points: Apply the basic principles and methods of mathematics, natural science, and engineering science to rationally optimize the experimental object according to the test and analysis results and give feasible suggestions and finite element analysis verification.
- 9 points: Apply the basic principles and methods of mathematics, natural science and engineering science; based on the test and analysis results, optimize the experimental object.
- 6 points: Under the guidance of teachers, apply the basic principles and methods of mathematics, natural science, and engineering science according to the test and analysis results and optimize the experimental object.

2) Temperature influence (6.5 points)

- 6.5 points: In the experimental report, the results were compared, and the temperature influence is summarized correctly.
- 5.2 points: In the experimental report, the results were compared, and how the temperature should affect the result is summarized.

3)Result analysis (21 points)

- 21points: In the experimental report, the finite element results and the test results can be compared and analyzed. The reason for the difference between the finite element and the test results is properly summarized.
- 19 points: In the experimental report, the finite element results and the test results can be compared and analyzed. Part of the reason for the finite element and the test results is properly summarized.
- 17 points: In the experimental report, the finite element results and the test results can be simply compared and analyzed.

### *2.4. The Teaching Method*

The comprehensive experimental teaching time of the finite element is 8 h. The experimental time of each part is shown in Table 2.

**Table 2.** The procedure of the experiment.

| | Experimental Procedure | Experimental Content | Time |
|---|---|---|---|
| 1 | Preview | Preview the method of stress-strain measurement, stress calculation method and finite element analysis software operation procedure. | 30 min |
| 2 | Introduce | Introduce the basic principles and methods of stress measurement and the principle of finite element analysis | 20 min |
| 3 | Interaction | Question and interact with the students in the difficulty of the experiment. | 10 min |
| 4 | Operation | Students survey and map the experiment components; Test the steps of the experiment; | 3 hrs |
| 5 | Analysis | The analysis of the experimental subject. | 3 hrs |
| 6 | Report submitting | Students analyze the experimental results and submit reports and instructors review the reports | 1hr |

## 3. Evaluation and Student Feedback

The outcome-based performance evaluation is an effective way of evaluating students' achievement and also gives effective feedback to make continuous improvements to the course. Therefore, the evaluation of the course objective achievements is an indispensable part of teaching after the experiment. We use two methods to evaluate the course objective achievement—the direct evaluation method based on student scores and the indirect evaluation method based on student questionnaires.

*3.1. Direct Evaluation Method*

As mentioned above, there are two objectives for this experimental project. These two objectives are supported by different experimental sessions as listed in Table 3. Objective 1 is supported by the above-mentioned Steps 1 and 3 with a total score of 25 points. Objective 2 is supported by Steps 2 and 4, with a total score of 75 points.

**Table 3.** Result comparisons.

| Step | Content | Course Objectives |
|---|---|---|
| 1：Structural mechanics analysis | 1、　Get the abstract structure<br>2、　Calculate the stress and strain using theoretical mechanics | 1 |
| 2：Experimental test | 1、　Link the experiment data collection system<br>2、　Install strain adapters and strain sensors<br>3、　Operate the experiment software | 2 |
| 3：The structure analysis by FE software | 1、　Abstract the three-dimension models<br>2、　Mesh the models<br>3、　Determine the boundary conditions and load<br>4、　Calculate the numerical result | 1 |
| 4：The result analysis and structure optimization | 1、　Compare the results of the three methods<br>2、　Optimize the structure design | 2 |

Achievement of course objectives (ACO) can be used for improving the teaching method, which can be calculated as follows:

$$ACO_j = \frac{\sum_{i=1}^{N} x_j(i)/N}{TS_j} \quad ACO_j = \frac{\sum_{i=1}^{N} x_j(i)/N}{TS_j} \tag{2}$$

where $N$ is the number of students and $x_j(i)$ is the score of the $i$th students in the $j$th curriculum objective. $TS_j$ is the total score of the jth curriculum objective.

In 2017, the number of students was 218, and the average score was 84.375 out of 100, where the average score of objective 1 was 22.825 and the average score of objective 2 was 61.55. In addition, we further analyzed students' grasp in all aspects. First, we divided the students into three parts according to their total scores—outstanding (scoring at least 90% of the points), excellent (scoring at least 85%of the points), good (scoring at least 80%of the points), and pass (scoring at least 60%of the points). Figure 3 shows the distribution of total points and the scores of the two course objectives. Figure 4a is the overall score distribution. We can see that some students still do not have high grades. We then further analyzed the achievement of the two objectives. We can see that the achievement level of Objective 1 is very high, indicating that the students have a very good command of the basic knowledge of mechanics. The achievement level of Objective 2 is relatively low.

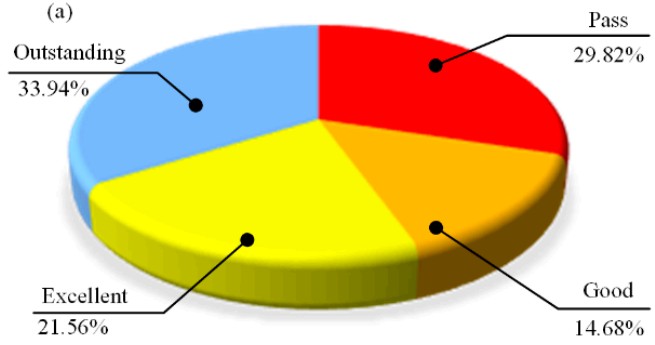

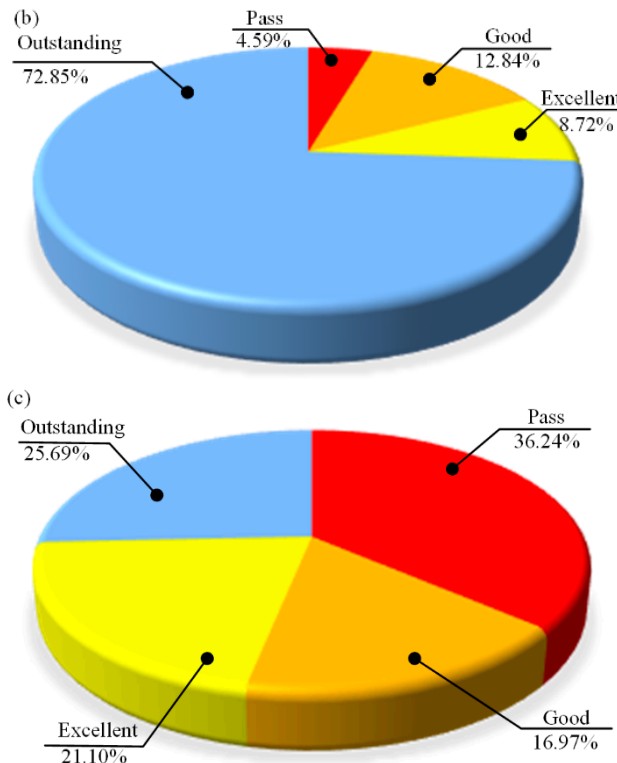

**Figure 3.** (**a**) The overall score distribution in 2017. (**b**) The score distribution of Objective 1 in 2017. (**c**) The score distribution of Objective 2 in 2017.

Further analysis shows that the students have achieved more than 90% of operating equipment and completing the experiment, indicating that the students have good practical ability. But the completion degree of structural optimization is the lowest, which means that students have problems in applying the results of finite element analysis to structural design. Through the above course outcome evaluation results, we made a course quality improvement plan, which was then conducted in 2018.

In 2018, the course quality improvement plan was carried out, and the achievements of the course objectives were evaluated again. We found that the average score of the students was 91.25, indicating a good teaching effect. The average score of objective 1 is 23.275 and the average of curriculum Objective 2 is 68.038. Figure 4 shows the distribution of total points and the achievement of two objectives in 2018. Then, we took the students located in the pass interval as the focus of attention and further analyzed their mastery of the knowledge. We found that they took the maximum stress point as the result of finite element analysis which leads to the inconsistency between the measurement and the result of the analysis. In the analysis of the calculation results, the consistency between the model, and the real object is neglected.

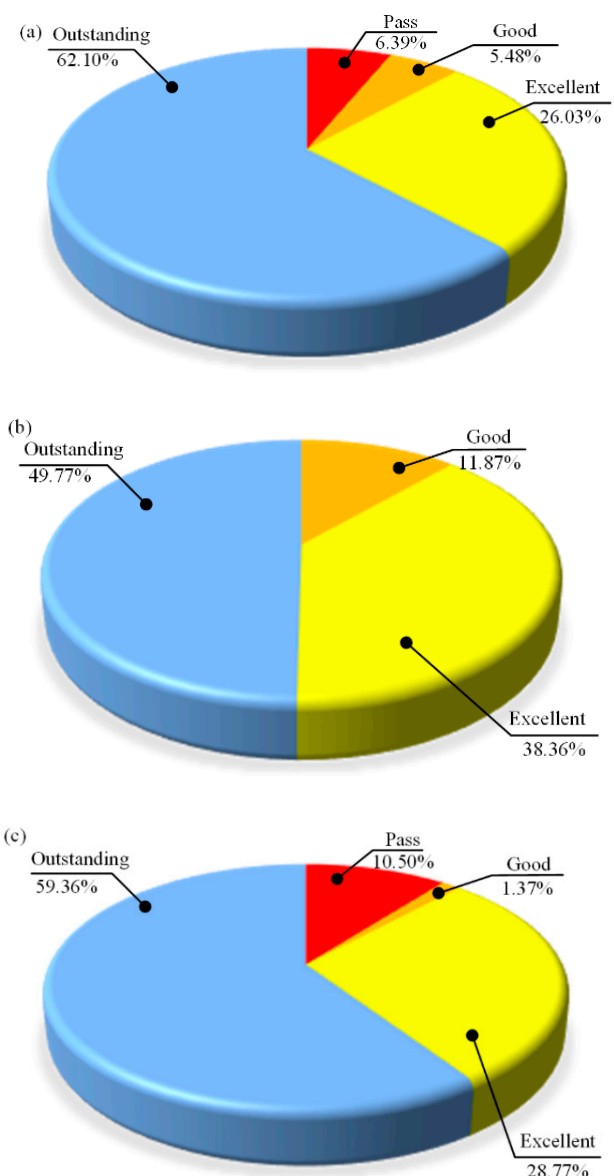

**Figure 4.** (**a**) The overall score distribution in 2018. (**b**) The score distribution of Objective 1 in 2018. (**c**) The score distribution of Objective 2 in 2018.

Figure 5 shows the course objective achievement in 2017 and 2018. We can see that the total achievement increased from 84% to 91%, where the achievement of Objective 2 clearly increased. The reason is that, during the teaching process of 2018, we emphasize the influence factors of stress measurement, and students can recognize the influence of temperature, selection of measuring circuit, and accuracy of the model on experimental results. This verifies the effectiveness of the improvement plan made in 2017.

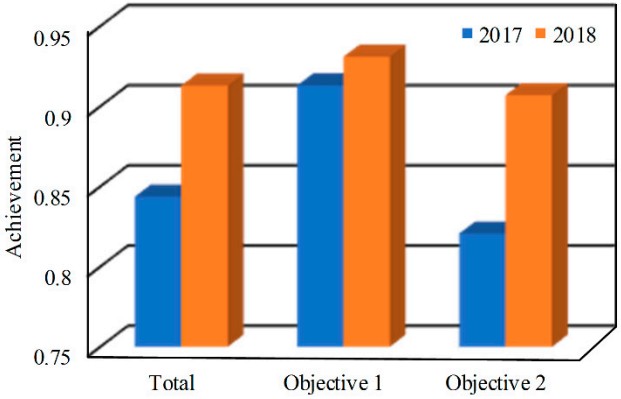

**Figure 5.** The course objective achievement in 2017 and 2018.

### *3.2. Indirect Evaluation Method*

Furthermore, we conducted a survey on students' learning satisfaction to analyze the course objective achievements indirectly in 2018. The participation of the survey is 83.56%.

In the questionnaire, we decomposed the course objectives into a number of questions, including a total of five questions in Objective 1 and a total of seven questions in Objective 2. The above scales are all based on the Likert five-point scale. The specific measurement topics are shown in Table 4.

**Table 4.** Satisfaction of experiment (N = 183).

| Variable | Topic | Agreement Level | | | | |
|---|---|---|---|---|---|---|
| | | 5 | 4 | 3 | 2 | 1 |
| **Objective 1** | ● I think that, through the experimental class, it is possible to achieve finite element calculation using the computer program. | 0.66 | 0.16 | 0.15 | 0.02 | 0.02 |
| | ● I am able to accurately and efficiently build a finite element model of the test subject. | 0.54 | 0.20 | 0.18 | 0.07 | 0.02 |
| | ● I can get reasonable results from the stress analysis of the experimental model through finite element analysis. | 0.59 | 0.21 | 0.13 | 0.05 | 0.02 |
| Objective 2 | ● I was able to select the appropriate circuit from the given stress-strain measurement circuit for measurement. | 0.46 | 0.28 | 0.18 | 0.07 | 0.02 |
| | ● I am able to standardize the installation of measuring sensors. | 0.26 | 0.28 | 0.25 | 0.00 | 0.05 |
| | ● I can standardize the experimental circuit. | 0.44 | 0.34 | 0.16 | 0.03 | 0.02 |
| | ● I can standardize the measurement of accurate and reliable experimental values. | 0.39 | 0.39 | 0.15 | 0.03 | 0.03 |
| | ● I was able to compare the experimental results with the finite element results and analyze the reasons for the gap. | 0.49 | 0.30 | 0.16 | 0.03 | 0.02 |
| | ● Based on the test and analysis results, I can reasonably optimize the experimental objects and give feasible suggestions and finite element analysis verification. | 0.44 | 0.33 | 0.16 | 0.03 | 0.03 |

The reliability and validity analysis using SPSS showed that the Cochran coefficients of Objective 1 and course goal 2 were 0.989 and 0.985, respectively, indicating that the scales have good internal consistency and high reliability. The KMO (Kaiser–Meyer–Olkin) of Objective 1 is 0.922, and the KMO of course goal 2 is 0.913, indicating that the scale structure is more effective.

The correlation between the three topics in Objective 1 is over 91%, and the correlation between

the six topics of Objective 2 is more than 90%, which proves the consistency between the data.

Figure 6 shows the agreement level distribution obtained from the student survey. We can find that most of the students believe the course objectives have been achieved. The average agreement level of Objective 1 is 4.3 and that of Objective 2 is 4.2. It should be noted that the average agreement level of Objective 1 is larger than that of Objective 2, which is consistent with the comparison between the achievements of the two course objectives obtained from the direct evaluation method.

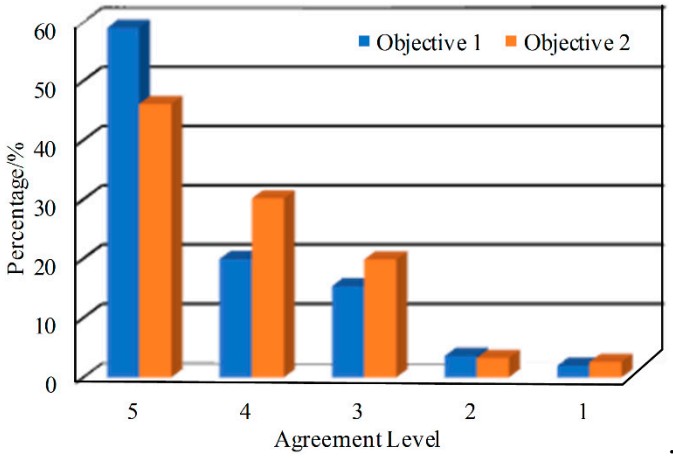

**Figure 6.** The agreement level distribution obtained from the students' survey.

Then the indirect course outcome evaluation was carried out based on student satisfaction data. The calculating method is similar to Equation 1 and the students' agreement level values are used instead of the students' score. Figure 7 shows the comparison of the results obtained by the direct evaluation method and indirect evaluation method. It is obvious that the achievement of Objective 1 is larger than that of Objective 2, with the two different evaluation results, suggesting the consistency and effectiveness of direct and indirect evaluation methods. However, we can also see that the objective achievement obtained by the direct evaluation method is slightly larger than that of the indirect evaluation method. We have further analyzed the reason for such a phenomenon. Most of the students encountered several difficulties during the experiments, although they have completed the task. Subjectively, their expectations are not high. This also indicates that the indirect course outcome evaluation results would be affected by subjective factors.

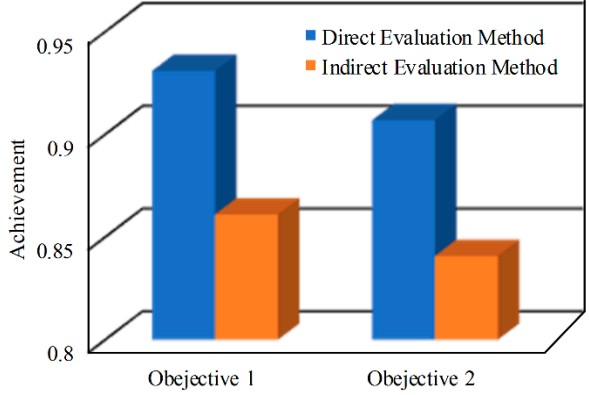

**Figure 7.** The companion of satisfaction and score in Objective 2.

## 4. Conclusions

This paper describes a typical case of setting up an experimental system based on engineering problems. The experiment project simulates the typical process of the application of finite element analysis in solving engineering problems, including engineering case modeling, system testing, and

structural optimization. Students can experience the process of finite element analysis in engineering. The student score is given according to the performance of students in each step of the experimental process. Then, the OBE-based course quality evaluation is carried out, and the results are used for the continuous improvement of the course.

**Author Contributions:** Conceptualization and writing—original draft preparation, Y. G.; investigation, L. Y.; methodology and funding acquisition, X. C.; supervision, L. Y.

**Funding:** This work was supported by the Research and Practice Project of emerging engineering education (Research on the Cultivation Demand and Countermeasure of Intelligent Engineering for New Engineering Science and Technology Talents) and the Shaanxi Association of Higher Education 2017 Higher Education Scientific Research Project (No. XGH17022).

**Conflicts of Interest:** The authors declare no conflict of interest.

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
