# Peer review of "An Engineering-Problem-Based Short Experiment Project on Finite Element Method for Undergraduate Students"

_education, doi:10.3390/educsci10020045_

Round 1

Reviewer 1 Report

In their paper entitled “Engineering-problem-based short experiment project on finite element method for undergraduate students,” the authors present a case for combining analysis, experiment, and simulation in a mechanics laboratory.  They show the results of one such experiment and then look at student achievement - both on a formal grading rubric based on how they ran the experiment and in informal interviews. 

In general, I conceptually agree with the authors that students need to understand mechanics from all three viewpoints and should be able to combine the viewpoints to ensure the simulation is valid and to optimize the resulting design.  I think the work presented here should be shared - although I have a few hesitations before recommending publication.

The grammar / writing needs some proofreading lines 52-63 are instructions for the authors - not material for the paper. Was the reason that 2018 did better than 2017 is because you told them?

My greatest concern with the work was that the authors are really only testing the student’s ability to follow directions - but not necessarily to think in a certain way (as evidenced by the setup, assignment, grading rubric, and discrepancy between test and interview assessments).  Why did the authors pick this particular hardware?  What does it mean to the students?  Does it have any relevance to them?  Why not let them pick a structure?  The grades seem to be based on how much of a recipe the students can execute on their own - why not instead grade them more on the depth and complexity of their analysis?

Author Response

  1. The grammar / writing needs some proofreading.

Response: We have revised the WHOLE manuscript carefully to improve the English writing.

  1. Lines 52-63 are instructions for the authors - not material for the paper

Response: We have deleted this part.

  1. Was the reason that 2018 did better than 2017 is because you told them?

Response: The improvement of the students’ achievements can be explained as follows: according to the course outcome evaluation results in 2017, we found the weak point of the students in the experiment project. Then, in 2018, we strengthened the training of the corresponding knowledge. We have revised relevant descriptions from Line 224 to 237 and from Line 252 to 257.

  1. My greatest concern with the work was that the authors are really only testing the student’s ability to follow directions - but not necessarily to think in a certain way (as evidenced by the setup, assignment, grading rubric, and discrepancy between test and interview assessments). Why did the authors pick this particular hardware? What does it mean to the students?  Does it have any relevance to them?  Why not let them pick a structure?  The grades seem to be based on how much of a recipe the students can execute on their own - why not instead grade them more on the depth and complexity of their analysis?

Response: We are not testing students' ability to follow the instructions. Actually, we provide information to students at the beginning of the experiment and require students to consult and learn methods from it. We always think the student could achieve the learning ability from this process.

  1. Why did the authors pick this particular hardware? What does it mean to the students? Does it have any relevance to them? Why not let them pick a structure?

Response: The hardware picked in the experiment is a typical mechanical structure, including cantilever beams, a typical mechanics structure. For the mechanical students, these structures are easy to understand and they can use the knowledge of previous courses on theoretical mechanics and material mechanics during experiments and calculations. Choosing the structure by students is not conducive to the implementation of the experimental part, because the processing cycle of these structures is relatively long.

  1. The grades seem to be based on how much of a recipe the students can execute on their own - why not instead grade them more on the depth and complexity of their analysis?

Response: The steps and grades are closely related. We divided the experiment based on the analysis steps of the finite element in the engineering case There also a quiz after the experiments to evaluate students’ analysis ability, depth and complexity of the analysis, which is an important part of the final score. Relevant descriptions were given from Line 174 to 196.

Reviewer 2 Report

The authors discuss an important issue in engineering education, namely practice ability and experimental projects. The manuscript is well written. I have some questions that would help to orientate the reader in the context and educational framework in which the experimental project is set.

How was the course taught before introducing the practical experiment with application of FEA? And what were the student results? What is the required knowledge of mathematics and physics for the course? In which study program is it taught? In which year of study is it taught?

Keywords: I would suggest not to use the abbreviation OBE but the whole term outcome-based education.

There are some minor language issues that should be corrected.

Author Response

  1. How was the course taught before introducing the practical experiment with application of FEA? And what were the student results? What is the required knowledge of mathematics and physics for the course? In which study program is it taught? In which year of study is it taught?

Response: Before introducing the experiment, the basic principle of finite element method and the operation of FEM software was taught to students. And students mastered the calculation method, finite element node division, and total rigid matrix composition. This course requires knowledge of linear algebra, higher mathematics, numerical analysis and theoretical mechanics. This course is taught in the first semester of the junior year for students majored in Mechanical Engineering. We have added relevant descriptions from Line 83 to 88.

  1. Keywords: I would suggest not to use the abbreviation OBE but the whole term outcome-based education.

Response: We have used the whole term outcome-based education as the keyword.

  1. There are some minor language issues that should be corrected.

Response: We have revised the WHOLE manuscript carefully to improve the English writing.